# Equity in Vaccine Trials for Higher Weight People? A Rapid Review of Weight-Related Inclusion and Exclusion Criteria for COVID-19 Clinical Trials

**DOI:** 10.3390/vaccines9121466

**Published:** 2021-12-11

**Authors:** Jessica Campbell, Juliet Sutherland, Danielle Bucknall, Lily O’Hara, Anita Heywood, Matthew Hobbs, Angela Ballantyne, Lesley Gray

**Affiliations:** 1Otago Medical School, University of Otago, Christchurch 8011, New Zealand; camje968@student.otago.ac.nz (J.C.); sutju965@student.otago.ac.nz (J.S.); bucda480@student.otago.ac.nz (D.B.); 2Department of Public Health, College of Health Sciences, QU Health, Qatar University, Doha P.O. Box 2713, Qatar; lohara@qu.edu.qa; 3School of Population Health, University of New South Wales, Sydney, NSW 2052, Australia; a.heywood@unsw.edu.au; 4School of Health Sciences, College of Education, Health and Human Development, University of Canterbury, Christchurch 8140, New Zealand; matt.hobbs@canterbury.ac.nz; 5GeoHealth Laboratory, University of Canterbury, Christchurch 8140, New Zealand; 6Department of Primary Health Care & General Practice, University of Otago, Wellington 6021, New Zealand; angela.ballantyne@otago.ac.nz; 7Centre for Biomedical Ethics, National University of Singapore, Singapore 117597, Singapore; 8Joint Centre for Disaster Research, Massey University, Wellington 6140, New Zealand

**Keywords:** vaccine, vaccination, COVID-19, higher weight, BMI, clinical trial, equity, inequity

## Abstract

Higher weight status, defined as body mass index (BMI) ≥ 30 kg/m^2^, is frequently described as a risk factor for severity and susceptibility to severe acute respiratory syndrome coronavirus 2 (SARS-CoV-2) disease (known as COVID-19). Therefore, study groups in COVID-19 vaccine trials should be representative of the weight spectrum across the global population. Appropriate subgroup analysis should be conducted to ensure equitable vaccine outcomes for higher weight people. In this study, inclusion and exclusion criteria of registered clinical trial protocols were reviewed to determine the proportion of trials including higher weight people, and the proportion of trials conducting subgroup analyses of efficacy by BMI. Eligibility criteria of 249 trial protocols (phase I, II, III and IV) were analysed; 51 protocols (20.5%) specified inclusion of BMI > 30, 73 (29.3%) specified exclusion of BMI > 30, and 125 (50.2%) did not specify whether BMI was an inclusion or exclusion criterion, or if BMI was included in any ‘health’ screenings or physical examinations during recruitment. Of the 58 protocols for trials in phase III and IV, only 2 (3.4%) indicated an intention to report subgroup analysis of vaccine efficacy by weight status. Higher weight people appear to be significantly under-represented in the majority of vaccine trials. This may result in reduced efficacy and acceptance of COVID-19 vaccines for higher weight people and exacerbation of health inequities within this population group. Explicit inclusion of higher weight people in COVID-19 vaccine trials is required to reduce health inequities.

## 1. Introduction

The development of severe acute respiratory syndrome coronavirus 2 (SARS-CoV-2) vaccines began early in the coronavirus disease (COVID-19) pandemic, with rapid movement from feasibility studies through to human clinical trials and administration of vaccines in several countries [1]. Global confidence in COVID-19 vaccination has been buoyed by large randomised controlled trials demonstrating high vaccine efficacy and excellent safety profiles [2,3]. For clinical trials to accurately assess the safety and efficacy of COVID-19 vaccination, they must exhibit sufficient participant diversity to be generalisable to the global population.

We define higher weight status as a body mass index (BMI) ≥ 30 kg/m^2^. In 2016, the World Health Organization (WHO) reported that 13% of the global population were of higher weight status [4]. In 2018, national surveys found that 42.4% of adults in the USA and 3.7% of adults in China had a BMI ≥ 30 [5,6,7]. Historically, higher weight status has been considered a comorbidity in the literature, and therefore used as the rationale to exclude people with higher weight from clinical research participation [8]. This exclusion erodes autonomy and reduces access to experimental interventions from which participants may derive a benefit [8]. Reduced participation opportunities have both ethical and safety implications for higher weight people. Exclusion of any marginalised group based on characteristics such as weight status, gender, ethnicity, disability, or age reduces the generalisability of research findings, and may lead to a scarcity of evidence to inform the treatment of people within these population groups [9]. Clinical uncertainty in the treatment of population groups excluded from research results in reduced quality of care, diagnostic overshadowing, increased frequency of adverse events, and poorer health outcomes [9].

Pharmacological research reinforces the necessity of including higher weight individuals in COVID-19 vaccine trials. Differences in pharmacodynamics across the weight spectrum suggest that research in lower weight people may not always be extrapolated to higher weight people [8,10,11,12]. For instance, research on hormonal medications such as the levonorgestrel emergency contraceptive (LNG EC) pill has demonstrated varying efficacy across the weight spectrum [10]. Pooled analysis from two large, multi-centre randomised controlled trials demonstrated that the pregnancy rate following the use of LNG EC increased 5.3-fold for the BMI 30–35 group and 3.4-fold for the BMI > 35 group [10]. It is notable that the mechanisms driving these pharmacodynamic differences are not well understood as trials typically exclude people with higher weight at the time of enrolment [10].

Research has shown variation in vaccine efficacy across the weight spectrum in both animal and human models in response to influenza, tetanus, rabies and Hepatitis A/B vaccinations [11,12,13,14,15,16]. Measures of initial response titres to influenza vaccination have been positively associated with higher weight [11]. Conversely, vaccination induced CD8+ T-cell populations and influenza vaccine antibody titres over time appear to be negatively associated with a higher weight [11]. Reduced antibody responses to vaccination may be secondary to mechanical factors, such as the delivery of lower relative dose of vaccine or reduced absorption from the site [16]. High circulating IL-6, representative of chronic low grade inflammation, has also been associated with reduced specific antibody responses to vaccination [16]. The association between weight status and chronic inflammation is complex and multifactorial [17]. The assertion that BMI or adiposity directly mediates inflammation overlooks evidence for the contribution of exposure to weight stigma, which causes chronic stress in the body, with demonstrable dysregulatory effects on the immune and cardiovascular systems [17,18,19], independent of actual body weight. Harms arising from weight stigma are exacerbated by medical marginalisation of higher weight people, including their exclusion from vaccine research and development.

To have confidence in the safety and efficacy of COVID-19 vaccines, trials must include participants representative of the distribution across the weight spectrum of the global population. Appropriate subgroup analysis by weight status should be performed so that we can appropriately titrate vaccine dose regimes, evaluate safety, and ensure equitable protection for higher weight people against COVID-19. This review assesses whether COVID-19 vaccine trials are inclusive of higher weight people. Of the trials that include higher weight participants, we quantify how many trials in phase III and IV intend to undertake subgroup analysis of vaccine efficacy.

## 2. Materials and Methods

This rapid review method is based on the Cochrane Rapid Reviews Methods Group interim guidance [20]. The review protocol and the results have been reported in accordance with the Preferred Reporting Items for Systematic Reviews and Meta-Analysis Protocols statement [21]. This review is registered with PROSPERO (CRD42020226573).

### 2.1. Included Studies

We included trials in any clinical phase, with a registered protocol that evaluated the safety, immunogenicity and/or efficacy, of a novel COVID-19 vaccine. We excluded any trial that investigated a therapy that did not generate active immunity, included people previously infected by COVID-19, evaluated the efficacy of a vaccine designed to protect against other pathogens e.g., Bacille Calmette-Guerin (BCG) in the prevention or treatment of COVID-19, and protocols written in a language other than English. Participants in included trials were adults, aged 18 years or older. Trials were included irrespective of the sex, gender and ethnicity of their participants. There were no restrictions on the setting, location or the country in which the trials were registered or conducted.

The interventions and exposures of interest are the participation of higher weight individuals in clinical trials of COVID-19 candidate vaccines registered with ClinicalTrials.gov, ISRCTN Register, the WHO official vaccine trial register, the ‘COVID-19 Vaccine Tracker’, and the Australia and New Zealand Clinical Trial Register between December 2020 and May 2021.

### 2.2. Outcome Measures

The number of vaccine trials including higher weight individuals as participants as a proportion of the total number of included COVID-19 vaccine trials. The number of higher weight individuals participating in trials (where recruitment is complete) as a proportion of the total number of included COVID-19 vaccine trial participants. Of trials where recruitment is complete, the proportion of trials planning to stratify their vaccine efficacy (VE) estimates by weight status, total numbers and percentages, and all periods of time and duration of follow-up.

### 2.3. Search Strategy

We performed a search of ClinicalTrials.gov, ISRCTN Register, the WHO official website for vaccine trials, the ‘COVID-19 Vaccine Tracker’ developed by the Vaccine Centre at the London School of Hygiene & Tropical Medicine, and the Australia and New Zealand Clinical Trial Register to identify clinical trials for vaccines in different countries irrespective of publication status, publication year, and language. Prior to final analysis, the search was repeated in May 2021. Details of the search strategy are outlined below in Table 1.

### 2.4. Data Extraction (Selection and Coding)

Selection of trial protocols and data extraction were performed independently by two reviewers (JC, LG). Each reviewer screened for relevance and eligibility as per the inclusion and exclusion criteria detailed above. Rayyan, a systematic review web-based application was utilised in “blind-on” mode whilst appraisal and selection of protocols for eligibility were performed. Excluded protocols were reviewed by a third reviewer (MH) to reduce the risk of missing eligible protocols. Disagreements were resolved in conversation and by consensus between the three reviewers (JC, LG & MH).

An initial pilot review of ten selected trial protocols using a tailored extraction form was undertaken. Differences in data extraction were discussed and calibration of the form was undertaken prior to data extraction from the remaining eligible protocols.

The calibrated data extraction form was used to collect the general study data (first author, year and language of trial protocol, year and place(s) of trial), trial number, trial status (recruiting, current, complete), trial design (study phase, level of blinding), trial purpose (dose finding, dose confirmation), participant characteristics (numbers, age range and mean (or median), health status), and vaccines (vaccine and placebo administered, antigen type/dose, adjuvant type/dose, vector type, dose regime).

Outcome data were extracted from the inclusion and exclusion criteria of the trial protocols. These data included whether the trial excluded participants with BMI > 30, the total number of participants and the number of participants with higher weight, and whether the trial intends to/has conducted analysis of vaccine efficacy (VE) by weight status. Where this information was not stated or was ambiguous, an email was sent to the protocol’s registered contact person for clarification.

## 3. Results

### 3.1. Search Results

A total of 489 protocols were identified and screened with 363 remaining after removing duplicates (*n* = 126). There were 249 trials meeting our inclusion criteria for rapid review (Figure 1).

### 3.2. Trial Characteristics

Trial Location: There were 59 countries registered as trial location (Appendix A). The majority of trial locations were in China (*n* = 44, 17.6% of trial registrations) and the United States of America (*n* = 36, 14.5% of trial registrations).

Trial Status: At the time of this review, few trials had moved beyond recruitment phase, with 221 (88.8%) trials active, not yet recruiting or in recruitment phases (Table 2). Only 21 trials (8.4%) had completed recruitment, and 14 of those (5.6% of the total) were completed trials. Vaccine platforms stated are included in Appendix A).

### 3.3. Participation of People with Higher Weight in COVID-19 Vaccine Trials

Of the 249 eligible trials, 51 (20.5%) included participants with a BMI > 30 across any of the four trial phases, 73 (29.3%) excluded participants with a BMI > 30 and 125 (50.2%) trials did not specify BMI as an inclusion or exclusion criterion in protocols or on request for further information (Table 3). Full details of included trial protocols are provided in Appendix A).

Of the 51 trials including participants with BMI > 30, the highest proportion were phase I trials (51.0%). Of the 73 trials excluding participants with BMI > 30, over half (58.9%) were phase I trials. BMI was less likely to be specified as an inclusion or exclusion criterion with advancing trial phase, with no phase IV trials specifying BMI as a criterion.

Of the 51 trials including participants with a BMI > 30, 5 (9.8%) trials excluded participants with a BMI > 32, 1 (2.0%) trial excluded participants with a BMI > 34, 29 (56.9%) trials excluded participants with a BMI > 35, 8 (15.7%) trials excluded participants with a BMI > 40, and 1 (2.0%) trial excluded participants with a BMI > 50. A further 7 (13.7%) trial protocols did not describe any upper BMI limit on participation (Table 4).

### 3.4. Participation of People with Higher Weight by Age

Most trials included participants with a wide range of ages. Adults aged > 65 years were included in 165 trials (66.3%), 119 trials (47.8%) included adults aged > 85 years, and 107 trials specified no upper age limit. The proportion of trials that included participants in each age group was similar for trials including or excluding participants with BMI > 30 (Table 5).

### 3.5. Participation of People with Higher Weight by Health Status

All trial protocols excluded participants who had previously been infected with COVID-19. Participant health status was not otherwise used as an inclusion or exclusion criterion for this review. Of the 249 trials that met inclusion criteria, 209 (83.9%) included participants categorised as “healthy” and 24 (9.6%) included participants categorised as “healthy” and “relatively healthy” with mild-to-moderate comorbidities or stable pre-existing medical conditions. Eight trials (3.2%) recruited participants with specific pre-existing medical conditions including chronic lymphocytic leukemia, liver disease, cancer, autoimmune disease, haemodialysis, and previous organ transplantation. Six trials (2.4%) included immunocompromised, immunosuppressed, and HIV positive people. Of these, 2 trials compared immunocompromised participants to healthy controls. Two trials (0.8%) included pregnant people (Table 6).

Participants with BMI > 30 were included in 40 trials (19.1%) requiring participants to be considered “healthy” by the clinical team as per the trial’s inclusion criteria (Table 6). By contrast, 9 trials (37.5%) that included participants with stable, mild-to-moderate pre-existing conditions included participants with BMI > 30. Some of these trials automatically assigned participants with BMI > 30 to the “comorbidities” arm of the trial.

### 3.6. Number of Higher Weight Individuals Enrolled in COVID-19 Vaccine Trials

There were insufficient participant data available or reported, at the time of this review, to be able to determine the number of higher weight individuals recruited into COVID-19 vaccine trials.

### 3.7. Intention to Analyse Vaccine Efficacy by Weight Status

There were 58 trials in phase III (*n* = 47) and phase IV (*n* = 11). Of these, only 2 trial protocols (3.4%) included an intention to stratify vaccine efficacy analysis by weight status.

## 4. Discussion

This rapid review assessed the extent to which COVID-19 vaccine trials were inclusive of higher weight people. Inclusion of higher weight people in COVID-19 vaccine trials was low, with only 51 (20.5%) of the trial protocols including participants with a BMI > 30. Most of these trials were in the early phases, with only 6% in phase III. Eligibility for vaccine trials for people with a BMI > 40 or 50 was particularly poor, with only 8 trial protocols including participants up to BMI of 40 and a single trial protocol including participants up to a BMI 50. Future trials should include people across the weight spectrum in proportion to their prevalence in the population.

Half (50.2%) of the trials did not specify BMI as an inclusion or exclusion criterion, but this does not guarantee the inclusion of higher weight people. Many of these trial protocols referred to the use of clinical judgement, physical examination, and “health questionnaires” to assess participants’ eligibility. It is feasible that BMI or weight status, while not explicitly mentioned as an exclusion criterion, formed part of the clinical assessment to determine participants’ suitability for participation. This is concerning as within a traditional weight-based medical paradigm, higher body weight is pathologised irrespective of health status [22,23,24]. The high proportion of trial protocols not specifying BMI within the inclusion or exclusion criteria makes it difficult to determine the true participation rates of people with higher weight in COVID-19 vaccine trials. Our findings are consistent with the eligibility and underreporting of inclusion or exclusion of higher weight participants in cancer treatment randomised controlled trials [8].

Exclusion of higher weight people in COVID-19 vaccine trials is high, with almost one third (29.3%) of all trial protocols explicitly excluding this significant population group that comprises 13% of the world’s population [4], and 42% of the population in the USA [6], where many of the vaccine trials are taking place.

Only 3.4% of phase III and IV trial protocols indicated an intention to undertake subgroup analysis of vaccine efficacy by weight status. Where protocols have not indicated an intention to perform subgroup analysis by weight status, and BMI data were collected throughout the recruitment process, retrospective analysis could be performed to establish vaccine efficacy and outcomes across the weight spectrum.

The impact of vaccination as a public health strategy relies on both vaccine effectiveness and vaccine acceptance. Vaccine effectiveness across the population requires the inclusion of people representative of the population in vaccine trials. This review demonstrates a significant gap with respect to higher weight people’s inclusion in COVID-19 vaccine trials. To ensure equitable outcomes for higher weight people, it is important that COVID-19 vaccine trials include participants that are representative of the full weight spectrum proportionate to the global population.

Vaccine acceptance requires trust, however medical marginalisation of higher weight people and weight stigma have historically eroded higher weight people’s trust in medicine and public health [25]. This reduces engagement of higher weight people in a range of preventive healthcare initiatives, such as cancer screening [26,27]. Exclusion of higher weight people from COVID-19 vaccine trials and incomplete analysis of vaccine efficacy may contribute to vaccine hesitancy for this significant population group.

Despite the relationship between weight status and COVID-19 outcomes remaining uncertain [28], pervasive assertions persist in media and medical publications that higher weight people are more likely to become infected with COVID-19, present with severe disease and utilise more healthcare resources [29,30]. Paradoxically, these assertions, and subsequent Government responses have resulted in the redirection of COVID-19 resources away from higher weight people across the Global North [31], further stigmatisation, and increased inequities. Reducing health inequities requires greater attention to the needs of people who have been marginalised by social and health care systems, not further exclusion as demonstrated in this review.

### Strengths and Limitations

A strength of this review is that it provides novel insights into the low level of explicit inclusion of higher weight people in COVID-19 vaccine trials. It is possible higher weight people were included in the 50.2% of trials that did not specify BMI as an inclusion or exclusion criterion in protocols or on request for further information. We recommend further reviews of participation and analysis of efficacy and safety profiles in higher weight participants in COVID-19 vaccine trials. A further strength of this review is its adherence to the interim guidance from the Cochrane Rapid Reviews Methods group, and utilisation of two reviewers to independently screen, extract and analyse the data. A limitation of this review is the low response rate to requests for further information from study authors. Additionally, we were unable to accurately quantify higher weight participation rates due to a lack of published or accessible demographic data sets. Few trials had completed recruitment at the time of undertaking this rapid review, with most in the active, not yet recruiting or recruiting phase. Of those trials with a completed recruitment phase, there were insufficient participant data to determine the proportion of higher weight individuals recruited into COVID-19 vaccine trials. We recommend a further review of trial participant data as they become available.

## 5. Conclusions

This rapid review has highlighted a distinct lack of COVID-19 vaccine trials that specifically included participants with higher weight. Moreover, there were very few trials that included participants with BMI > 40 or BMI > 50. Future research should prioritise equitable inclusion of people with higher body weight in COVID-19 vaccine trials. Omission or under-representation of participants with higher weight may result in poorer vaccine coverage for people with higher weight and contribute to greater health inequities.

## Figures and Tables

**Figure 1 vaccines-09-01466-f001:**
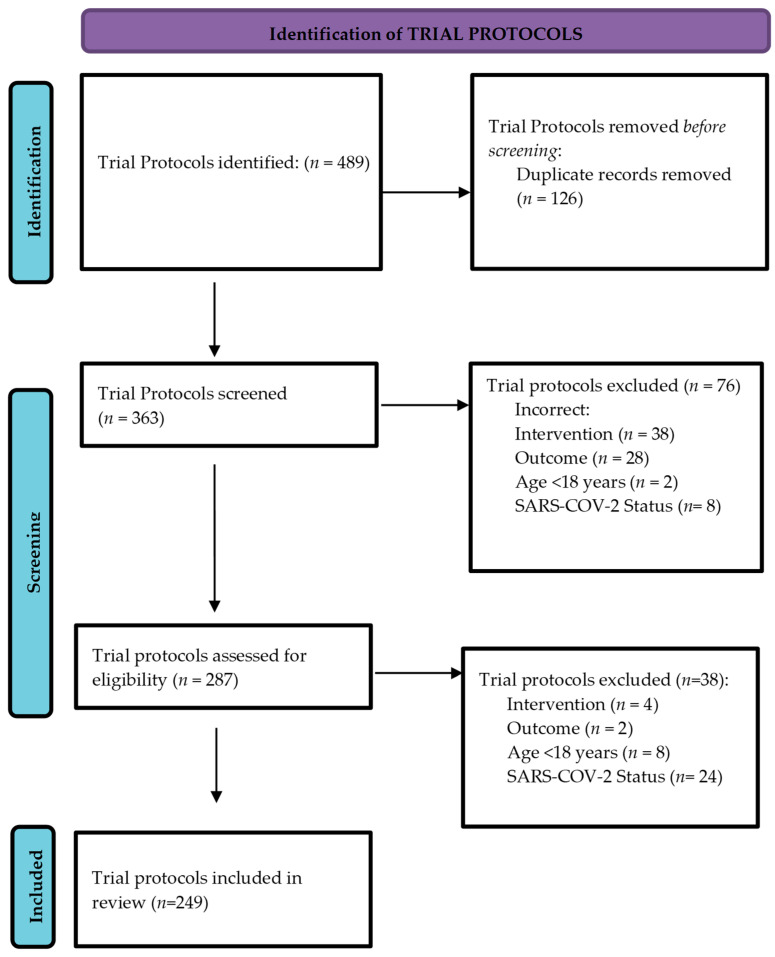
Identification of protocols for inclusion in review.

**Table 1 vaccines-09-01466-t001:** Search strategy, inclusion and exclusion criteria for rapid review.

Search Strategy		
**Condition or disease:** COVID OR Covid-19 OR SARS-Cov-2 OR Coronavirus OR Corona Virus OR nCOV OR novel coronavirus OR severe acute respiratory syndrome coronavirus 2 **Intervention/Treatment:** vaccine OR vaccines OR vaccination OR Vaccinations OR immunization OR immunizations OR inoculation OR inoculations **Study Type:** Interventional (Clinical Trials) **Applied filters:** recruiting, not yet recruiting, active not recruiting, completed, enrolling by invitation, suspended, terminated, withdrawn, unknown status		
	**Inclusion**	**Exclusion**
**Format**	Registered Clinical Trial Protocol, in any clinical phase	In-vitro studies, animal studies, non-human trials
**Intervention**	Trials of a novel COVID-19 vaccine evaluating efficacy, safety and/or immunogenicity	Trials evaluating therapies that do not generate active immunity Trials evaluating the efficacy of vaccines designed to protect against other pathogens (e.g., BCG vaccine)
**SARS-Cov-2 Status**	Trials assessing prior history of and exposure to COVID-19, or sero-positivity for COVID-19	Trials including people infected by COVID-19
**Age**	≥18 years	<18 years
**Language**	English language	Non-English language
**Location**	Any	NA

**Table 2 vaccines-09-01466-t002:** Eligible COVID-19 vaccine trial status at time of review.

Trial Status	
Status at Time of Review	Number of Trials *n* (%)
Terminated/Withdrawn/Suspended	5 (2.01)
Active, not-recruiting	114 (45.78)
Recruiting	107 (42.97)
Recruitment complete	7 (2.81)
Completed	14 (5.62)
Unknown	2 (0.80)
Total	249

**Table 3 vaccines-09-01466-t003:** BMI > 30 participation in COVID-19 vaccine trials by trial phase.

Trial Phase	Total Number of Trials *n* (% of Total Trials)	Trials Including BMI > 30 *n* (% of Phase)	Trials Excluding BMI > 30 *n* (% of Phase)	Trials with BMI Not Specified *n* (% of Phase)
I	80 (32.1)	26 (32.5)	43 (53.8)	11 (13.8)
I/II	59 (23.7)	14 (23.7)	22 (37.3)	23 (39.0)
II	29 (11.6)	6 (20.7)	6 (20.7)	17 (58.6)
II/III	14 (5.6)	2 (14.3)	0 (0.00)	12 (85.7)
III	47 (18.9)	3 (6.4)	2 (4.3)	42 (91.3)
IV	11 (4.4)	0 (0.0)	0 (0.0)	11 (100.0)
Not Specified	9 (3.6)	0 (0.0)	0 (0.0)	9 (100.0)
Total	249 (100)	51 (20.5)	73 (29.3)	125 (50.2)

**Table 4 vaccines-09-01466-t004:** BMI upper limits in trials with BMI > 30 by trial phase.

Trial Phase	Number of Trials Including BMI > 30	BMI Upper Limit					
		BMI = 32 *n* (% of Phase)	BMI = 34 *n* (% of Phase)	BMI = 35 *n* (% of Phase)	BMI = 40 *n* (% of Phase)	BMI = 50 *n* (% of Phase)	Upper Limit Not Stated *n* (% of Phase)
Phase I	26	3 (11.5)	1 (3.8)	15 (57.7)	4 (15.4)	1 (3.8)	2 (7.7)
Phase I/II	14	0 (0.0)	0 (0.0)	10 (71.4)	3 (21.4)	0 (0.0)	1 (7.14)
Phase II	6	2 (33.3)	0 (0.0)	3 (50.0)	1 (16.7)	0 (0.0)	0 (0.0)
Phase II/III	2	0 (0.0)	0 (0.0)	0 (0.0)	0 (0.0)	0 (0.0)	2 (100.0)
Phase III	3	0 (0.0)	0 (0.0)	1 (33.3)	0 (0.0)	0 (0.0)	2 (66.7)
Phase IV	0	0 (0.0)	0 (0.0)	0 (0.0)	0 (0.0)	0 (0.0)	0 (0.0)
Not Specified	0	0 (0.0)	0 (0.0)	0 (0.0)	0 (0.0)	0 (0.0)	0 (0.0)
Total Trials	51	5 (9.8)	1 (2.0)	29 (56.9)	8 (15.7)	1 (2.0)	7 (13.7)

**Table 5 vaccines-09-01466-t005:** BMI >30 inclusion and exclusion by age of participants in COVID-19 vaccine trials.

Age of Trial Participants	Number of Trials *n* (% of *n* = 249)	Trials Including BMI > 30 *n* (% of Age Group)	Trials Excluding BMI > 30 *n* (% of Age Group)	Trials with BMI Not Specified *n* (% of Age Group)
Age limit ≤45 years	6 (2.4)	2 (33.3)	2 (33.3)	2 (33.3)
Includes adults >45 years	243 (97.5)	49 (20.2)	71 (29.2)	123 (50.6)
Includes adults >65 years	165 (66.3)	34 (20.6)	32 (19.4)	99 (60.0)
Includes adults >85 years	119 (47.8)	16 (13.4)	18 (15.1)	82 (68.9)
No upper age limit specified	107 (43.0)	14 (13.1)	21 (19.6)	72 (67.3)

**Table 6 vaccines-09-01466-t006:** Inclusion and exclusion by health status of participants with BMI > 30 in COVID-19 vaccine trials.

Health Status of Trial Participants	Total *n* (% of Total Trials)	Trials Including BMI > 30 *n* (% of Health Status)	Trials Excluding BMI > 30 *n* (% of Health Status)	Trials with BMI Not Specified *n* (% of Health Status)
Healthy *	209 (83.9)	40 (19.1)	70 (33.5)	99 (47.4)
Healthy plus mild/moderate comorbidities *	24 (9.6)	9 (37.5)	3 (12.5)	12 (50)
Immunocompromised, not otherwise specified	4 (1.6)	0 (0.0)	0 (0.0)	4 (100.0)
Healthy * plus immunocompromised	2 (0.8)	2 (100.0)	0 (0.0)	0 (0.0)
Specific medical condition (CLL, liver disease, cancer, haemodialysis, and transplant patients)	8 (3.2)	0 (0.0)	0 (0.0)	8 (100.0)
Pregnant	2 (0.8)	0 (0.0)	0 (0.0)	2 (100.0)
Did not specify	0 (0.0)	0 (0.0)	0 (0.0)	0 (0.0)
Total	249 (100)	51 (20.5)	73 (29.3)	125 (50.2)

* as assessed by clinical team in the trial.

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
