# Peer review of "Equity in Vaccine Trials for Higher Weight People? A Rapid Review of Weight-Related Inclusion and Exclusion Criteria for COVID-19 Clinical Trials"

_vaccines, 2021, doi:10.3390/vaccines9121466_

Round 1

Reviewer 1 Report

1. the paper intiteled "Equity in vaccine trials for higher weight people? A rapid review of weight-related inclusion and exclusion criteria for COVID-19 clinical trials" by Campbell J et al report interesting informations, I suggest to shortened the discussion section

2. The main question addressed by the research is evaluating vaccine outcomes for higher weight people in Trial protocols.

3. Topic is original and relevant in the field. It is important to know  the characteristics of the patients included in the clinical studies to know the validity of the results on the real population.

4. The authors highlights the lack of data on the efficacy of vaccines in people with higher weight compared with other published material.

5. specific improvements regarding the methodology: it would also be  interesting to differentiate according to the type of vaccine  administered. 

6. The conclusions are consistent with the evidence and arguments presented and they address the main question posed.

Author Response

We thank Reviewer 1 for their timely review and considered comments. We hope the reviewer agrees that this has contributed to a much improved manuscript, now attached. We also provide our response to reviewer comments below: 

MDPI VACCINES Reviewer Comments

Author Responses

#Reviewer 1

the paper intiteled "Equity in vaccine trials for higher weight people? A rapid review of weight-related inclusion and exclusion criteria for COVID-19 clinical trials" by Campbell J et al report interesting informations, I suggest to shortened the discussion section.

We thank Reviewer 1 for their positive feedback.  We have shortened the discussion section as suggested.

Additional points submitted by reviewer after initial notification:

The main question addressed by the research is evaluating vaccine outcomes for higher weight people in Trial protocols.

We thank the reviewer for the additional comments, however this manuscript is not about research evaluating vaccine outcomes for higher weight people.  The review we undertook examines inclusion of people at higher weight in vaccine trials (inclusion and exclusion criteria).

Topic is original and relevant in the field. It is important to know the characteristics of the patients included in the clinical studies to know the validity of the results on the real population.

We thank the reviewer for the additional comments, however this manuscript is not about research evaluating vaccine outcomes for higher weight people.  The review we undertook examines inclusion of people at higher weight in vaccine trials (inclusion and exclusion criteria) and we are not reporting on Trial results.

The authors highlights the lack of data on the efficacy of vaccines in people with higher weight compared with other published material.

Thank you.

specific improvements regarding the methodology: it would also be interesting to differentiate according to the type of vaccine administered.

We thank the reviewer for the additional comments, however the review we undertook examines inclusion of people at higher weight in vaccine trials (inclusion and exclusion criteria).  All vaccines related to COVID-19 and each trial was evaluating their own vaccine. See also sections 3.6 and 3.7 of the manuscript.

The conclusions are consistent with the evidence and arguments presented and they address the main question posed.

Thank you.

Reviewer 2 Report

This is a rather stringent rapid review that highlighted a distinct lack of COVID-19 vaccine trials that specifically detailed inclusion of participants with higher weight - and as such is of high relevance. I suggest the acceptance of this paper, with certain smaller issues that have to be addressed before publication in the journal Vaccines.

All abbreviations should be initially stated in full. Although well-established now, this is also valid for SARS-CoV-2, COVID-19, BMI, etc. Furthermore, the evidence for the claim "Experiences such as weight stigma, chronic body shame, and medical marginalisation cause prolonged stress in the body with demonstrable effects on both the immune and cardiovascular system" is rather weak, so I suggest rephrasing this part.

The Methods section of the paper should aim to explain the risk of bias (ROB) and ROB-2 Tool in more depth, regardless of provided citation. Considering the tone of the Introduction section, the Discussion section should also emphasize how groups who have faced stigma and marginalization in health care (which inclueds individuals with obesity) deserve increased attention in our efforts to convey trust in current COVID‐19 vaccination efforts (and beyond).

Author Response

We thank Reviewer 2 for their timely review and considered comments. We hope the reviewer agrees that this has contributed to a much improved manuscript, now attached. We also provide our response to reviewer comments below: 

MDPI VACCINES Reviewer Comments

Author Responses

#Reviewer 2

This is a rather stringent rapid review that highlighted a distinct lack of COVID-19 vaccine trials that specifically detailed inclusion of participants with higher weight - and as such is of high relevance. I suggest the acceptance of this paper, with certain smaller issues that have to be addressed before publication in the journal Vaccines.

We thank Reviewer 2 for their positive feedback.  We have attended to the smaller issues identified, as below:

All abbreviations should be initially stated in full. Although well-established now, this is also valid for SARS-CoV-2, COVID-19, BMI, etc.

We have attended to the abbreviations, thank you.

Furthermore, the evidence for the claim "Experiences such as weight stigma, chronic body shame, and medical marginalisation cause prolonged stress in the body with demonstrable effects on both the immune and cardiovascular system" is rather weak, so I suggest rephrasing this part.

Thank you for this comment. This section has been revised to highlight the contribution of weight stigma to systems dysregulation, and how these harms are exacerbated by medical marginalisation such as exclusion from vaccine trials.

The Methods section of the paper should aim to explain the risk of bias (ROB) and ROB-2 Tool in more depth, regardless of provided citation.

The Risk of Bias results are not reported in the manuscript and so the reference to the ROB has been deleted from the methods section.

Considering the tone of the Introduction section, the Discussion section should also emphasize how groups who have faced stigma and marginalization in health care (which inclueds individuals with obesity) deserve increased attention in our efforts to convey trust in current COVID‐19 vaccination efforts (and beyond).

This is an excellent point and we have revised the discussion to include it.